# I did not scream. i could not; i was terrified. i just followed them. . .i blocked my mind. then they all raped me: A narrative inquiry on the onset of tonic immobility among women rape victims in Nigeria

**Dooshima Dorothy Gbahabo** *, **Sinegugu Evidence Duma**

Discipline of Nursing, School of Nursing & Public Health, College of Health Sciences, University of KwaZulu-Natal, Howard College Campus, Durban, South Africa

* dooshima.gbahabo@gmail.com

## Abstract

Tonic immobility (TI) is a common uncontrollable autonomic mammalian response to an extremely fearful situation. It is one of the most immediate devastating consequences of rape and remains poorly understood. While controversies over its definition persist among researchers, this also reflects on the care for and support to victims. The study aimed to explore and describe the onset of TI and the meaning attached to the experience among women raped victims in Nigeria. The study design was the qualitative narrative inquiry approach. Criterion and purposive sampling were conducted across four post-rape care facilities in Lagos, Nigeria, to recruit 13 participants. In-depth face-to-face interviews were conducted using a semi-structured interview guide to generate data that were thematically analysed. The findings of the study revealed five themes relating to the onset of TI as follows: the first two focused on the experience of TI: (1) the onset of TI prior to rape due to perceived imminent danger, (2) the onset of TI as a self-protection mechanism from further harm) while the last three relate to the meaning-making of the experience of TI (3) self-loathing as a meaning attached to TI, (4) suicidal ideations as a meaning attached to TI, and (5) divine intervention as a meaning attached to TI. Conclusion: The findings underscore the experiences and meanings that participants attach to TI following rape. There is a strong likelihood that tonic-immobility is not an uncommon experience amongst rape victims, but that in the absence of research, specialized care on the condition, and its associated consequences will haunt many women, affecting their psychological well-being and their entire quality of life. Describing the phenomenon as it is experienced by the participants is critical because understanding the condition is the first step toward effective appropriate management.

**Data Availability Statement:** All relevant data are within the paper.

**Funding:** The authors received no specific funding for this researh.

**Competing interests:** The authors have declared no competing interests exist.

## Introduction

More than 250,000 rape cases are registered yearly worldwide [1,2] this highlights rape as a global public health challenge in many countries because of its dire consequences [3,4]. In Nigeria, there is sparse national information with only two out of 40 rape cases being reported to the police [5,6]. Obviously, less research has been dedicated to in-depth studies of rape incidence in the country, hence sparse provable data on rape cases, [6,7] Available data from different states in the country had shown a decrease in rape prevalence ranging from a 15% decrease in Ibadan, Oyo State [8], 13.8% in Maiduguri, Borno State [9], and 0.8% in Lagos, Lagos State[5]. However, despite declining state rape data, a 2017 study on the experience of violence among female health workers in Nigeria by [10] showed that sexual abuse (SA) (71.7%) was most common form and generally about 90% of all reported SA cases were related to rape [11]. An immediate and devastating consequence of rape is tonic immobility (TI) of which [4], reported that out of 298 women raped, 70% experienced it.

TI, an uncontrollable autonomic mammalian response to an extremely fearful situation, is one of the most immediate consequences of rape, which is triggered when fighting the rapist does not appear conceivable and there are limited other means for self-protection [4,12–15]. TI is a product of the defence cascade that involves stress, fear, the threat to life, and is one of four outcomes that manifest when the body system is highly stressed: fight, flight, freeze, and TI [16]. The onset of TI manifests itself as a reversible catatonic-like state with muscle hyper- or hypo-tonicity; tremors, trembling or shaking; the inability to vocalise or control the muscles and movement, feeling of numbness, cold, detachment from self and the immediate environment; and relative unresponsiveness to external stimuli in a context of the inescapable threat [4,14,17]. Other signs of TI include intense fright expressed at a later recant with phrases such as "I was scared stiff" and "I was frozen with fear" [18]. According to [19], women are more likely to report such experience if they were sexually abused as children. In addition, while TI may shield a victim from further physical injuries and the severity of an attack, its occurrence due to SA trauma has been linked to the development of post-traumatic stress disorder (PTSD). Extant literature associates TI with survivors' reporting personal experiences of psychological nature such as humiliation, guilt, and self-blame from the perceived failure to protect themselves during the rape incident [14,20,21].

Although TI has been identified as the most immediate consequence of rape, it has not yet been fully explored in humans, being an emerging area of research interest globally, and in Africa specifically [4,14,22]. This may explain why very little is known about its occurrence also, among Nigerian women victims of SA [23], or why it is not understood by the public, the legal system and health practitioners who deal with survivors seeking legal and health care. Such lack of understanding of TI by the major stakeholders dealing with victims and survivors of SA may lead to the incapability to identify and address the immediate and associated long-term consequences, resulting in unacceptable health and legal outcomes for victims and survivors.

Knowledge about TI, and the recognition of its onset and manifestations, are critical for all health professionals and first responders, especially the clinical forensic nurses and doctors who are responsible for providing appropriate and immediate nursing management and clinical forensic medical to the victims and survivors of SA. The victims and survivors, their families, and the police also need to know and understand TI, its onset, and various its clinical manifestations. Such knowledge may reduce self-loathing among those who experience it from SA or are blamed by others for having not protected themselves during the incident.

There are however different thoughts about what TI is among scientists [12,19,24–26]), including opposing views regarding its nature in rape victims. For instance, while [27,28] refer

to the phase of mental paralysis during or immediately after rape as TI and similar to a freeze response,[29] argue that it occurs early in the encounter stage of a defensive reflex. On the other hand, for [30], the onset of TI is shown by a rape-induced paralysis, the two actions denote the same phenomenon. This lack of clarity as a standard for TI may have implications in decision-making towards developing appropriate intervention algorithms [31]. It was stressed by [16] that understanding these concepts leads to tangible means of preventing mal-adaptive behaviours as consequences.

The aftermath of the devastating effect of rape, may often lead victims or survivors in the attempt to find meaning to their predicament [32,33]. Meaning making manages hurtful events that result in uncertainty in a victim's global and situational meaning arrangements [34]. However, if the global meaning (beliefs why bad things happen, values) of victims is not consistent with the situational meaning (one's circumstances and how those circumstances are understood), this will result in distress and as such, meanings will be made to cope with the distress [35]. According to Park and other authors [36–38], meaning making aims to restore disrupted global views by victims engaging in mental processes to develop new and acceptable ways of understanding a stressful situation that are consistent with their held beliefs or values or by changing them to favourable ones, an outcome of this is called 'meaning made' [38]. But if the stressor and situational meaning are consistent, there will be no need for meaning making. This study was therefore undertaken to explore and describe the onset of TI and the meaning participants made of their experiences of it.

## Materials and methods

### Research design

As a qualitative research design, the narrative inquiry (NI) approach was used due to its conversational nature in data collection, and its ability to allow the participants to easily tell their stories on the onset of TI and share their experiences with the researcher. This approach accorded the first author an 'insider view' for a deeper understanding of TI's onset from the participants' perspectives [39].

### Study setting

The study was conducted in three purposefully selected government general hospitals' post-rape centres and one registered non-profit, non-government organisation (NGO) facility that provide care and support services to survivors of sexual assault in Lagos, Nigeria. Each of the selected government hospital was reported to be receiving an average of seven adult rape victims per month, while the NGO, which also provided services to post rape cases, recorded approximately 30 rape victims monthly, including adults and children. The reasons for more victims attending the NGO facility could be due to the facility being better equipped with full spectrum of services required and its enhanced privacy.

### Sampling and sampling size

Criterion-based, deliberate, and purposive sampling strategies were applied to select and recruit 13 adult women who self-reported as having been raped within a period of 12 weeks from the start of study at the identified health facilities. The sample size was attained through data saturation, meaning that no new knowledge was obtained from the preliminary analyses of data after the 13th participant, with further recruitment of participants for data collection deemed redundant [40]. This sample size was considered adequate and acceptable for this

qualitative study, as the focus was to have a deeper understanding regarding the onset of TI from the participants' perspectives rather than to generalise the findings [41,42].

## Inclusion and exclusion criteria

Potential participants were included in the study if they were aged 18 years or older, had self-reported to have been raped within 12 weeks of the start of the study on 1 December 2018, and had experienced at least one or more manifestations of the onset of TI listed in the study's information sheet. The manifestations included the inability to control their muscles and movement, call out or scream; numbness; fast heartbeat; rapid breathing, and fear for their life [43–45]. They had to understand and communicate in English or the Nigerian Pidgin English language, as Nigeria has over 350 languages, with Yoruba, Hausa, Igbo, English, and Pidgin English being official. They needed to have access to a functional mobile phone for sustained communication during data collection and for member checking [46,47]. Potential participants who reported or were known to have mental health problems were excluded from participating to avoid exacerbating their condition due to the sensitive nature of the research questions [48], in compliance with the principle of doing no harm [49].

## Recruitment of study participants

Participants recruitment began after the ethical clearances were obtained and were conducted by the first author with the assistance of trained counsellors who were assigned in each facility (Facility Focal Person, (FFP)). They liaised with the first author once potential participants were identified. Each FFP was trained in his/her facility on the study's purpose, the importance of confidentiality, and the eligibility criteria to enable them assist with recruitment, inform participants of the study and ask for their willingness to participate in the research. Women who were willing to participate were linked by phone to the first author for further interaction. A preliminary interview was held to confirm their experience of TI before recruitment into the study and to initiate a trust relationship between the participants and the first author [50,51]. Dates, times, and locations for each in-depth individual interview was negotiated with each participant. In line with the principles of NI, the first author sustained the developed relationship and interest with the 13 participants [52]. After each in-depth interview, the women were reimbursed for their travel costs [22,53].

## Pilot study

Two participants were purposefully recruited, one each from a government hospitals' post-rape centre and the NGO for a pilot study, which was conducted in December 2018, using the study's pre-determined inclusion criteria. The pilot study aimed to assess the adequacy of the semi-structured interview guide that was designed by the authors to stimulate the desired narratives from the participants. The analysed pilot data validated the appropriateness, simplicity, and adequacy of the in-depth interview questions in eliciting the necessary data. A modification to the in-depth interview guide was suggested by a counsellor at the NGO such as the word 'rapist' replaced with 'assailant'. The findings from the pilot study were included with the data from the main study and did not contaminate the results in any way [53].

## Data collection

The data collection for the main study started late in December 2018 and was completed at the end of August 2019, with interviews conducted in English and/or Nigerian Pidgin in settings chosen by each participant for their comfort and safety [54]. These included the NGO post-

rape care facility (N = 1); the first authors' vehicle in a public parking lot (N = 1) and across the three government general hospitals (N = 11) in Lagos on weekdays.

Despite the ethical approvals from the ethics committees, a particular gatekeeper requested for a 'Disclaimer' document which was provided by the University of KwaZulu-Natal with which a go ahead was then given for data collection.

Informed consent was sought from each willing participant because this point signified a high level of willingness to go through with the in-depth interviews. Thus, after also reminding them of opt out option, each participant was given a prepared consent form to endorse her consent.

The participants demographic details were obtained (age, education, type of rape) using a short questionnaire. A semi-structured interview guide was used, and the main question was: "*When you think back, at what period did you first experience all the bodily reactions you earlier identified?*". Following the NI approach, each participant was given time to think, reflect and gather her thoughts before responding to the question. This was followed by a few probing questions depending on a participant's narration. During the in-depth face to face interview of approximately 60 minutes' duration, each participant narrated her story of how she experienced the manifestation of TI but only as much as she wanted [51].

### Data management and data analysis

The participants were identified by the serial numbers they were given on recruitment, for example, participant P1 and on to the last participant as P13, and then their age which was captured in ranges. These were used to code their data, audio files, transcripts, manuscripts, and folders to ensure anonymity and confidentiality [55], with the facilities' initials being as folder acronyms. The digital data was stored on a password-protected computer, all of which will be destroyed after five years in accordance with the University of KwaZulu- Natal's policy.

Braun and Clarke's [56] six steps of thematic analysis were used to manually analyse the narrative data obtained from the participants. Firstly, the first author transcribed the audio recordings verbatim, and cleaned the text by conducting quality checking against the original recording. This entailed listening to the audiotapes and noting first impressions before typing out the transcripts [57]. Each participant's narrative was read reiteratively and inductively to ensure a thorough grasp of the data while noting patterns of interest across all their data [58]. Secondly, notable features were coded to help organise the data and describe the nature of the content, with reiterative reflection enabling new insights to be observed on repeat reading. Thirdly, a list of the codes was created and examined to identify patterns within them, and to begin generating likely themes, which were grouped according to their relationships and similarities. Fourthly, the developed themes were reviewed and modified, the relevant data being assigned to each. Fifthly, the emerging themes were defined and pegged with meanings to indicate their significance [59]. The dossier of themes and related quotes from the raw data was shared with the second author for confirmation and verification. Both authors reviewed, redefined, and renamed the themes by aligning the meanings of the raw data with the characteristics of the onset of TI [60]. The sixth and final step involved the production of this scholarly report of the analysis [56].

### Ethical considerations

Ethical clearances were granted by the Biomedical Research Ethics Committees of the University of KwaZulu-Natal (UKZN), Durban, South Africa (BE 402/18), and the Lagos University Teaching Hospital Health Ethics Research Committee (AD/DCST/HERC/APP/2448). Gate keepers' consent was obtained from the Director of the Lagos State Primary Health Care

Board (LS/PHCB/MS/ 1128/VOLIV/073), and the Director of the Lagos State Health Services Commission (LSHSC/ DNS/RESEARCH/VOL.III/41) as Coordinating agencies of the facilities from where participants were recruited. Verbal approvals to collect data for the study was given by each of Medical Directors of the three governmental health facilities and the one NGO included in the study on presentation of the approvals from their coordinating agencies.

The set of ethical principles intended to protect the human subjects, [49] were applied throughout the data collection and analysis processes, and the following five ethical considerations in qualitative research were applied [61]. Firstly, a relationship was developed with the participants and trust built to enable their richer stories to be gained. Secondly, informed consent was upheld by explaining the research process, as contained in the health information sheet, in simple English and Nigerian Pidgin English. Participants were encouraged to ask questions for clarification and to their satisfaction after which their consents were sought and obtained through individualized signed consent or thumb printed predesigned consent forms. Each participant informed that that they could opt out of the research at any point without any consequences. Thirdly, anonymity was upheld by protecting their identities and their documents using pseudonyms. Fourthly, to uphold beneficence, negotiated arrangements were respected. To avoid re-traumatising the participants, the research questions had been first pilot tested for language suitability. Fifthly, the research findings were presented with honesty integrity, respect for participants and it was a re-echo of their voices. The first author was reflexive at every stage of the research process to avoid influencing the evidence supports to any conclusion when it did not do so. This was exhibited in the first author's mindfulness of her role in the research process and in the deliberate efforts that reduced subjectivity. The procedure of obtaining consent was approved by the ethics committees of the University of KwaZulu-Natal, Durban, South Africa, and the University of Lagos, Nigeria. Additional information regarding ethical, cultural, and scientific considerations specific to inclusivity in global research has been included as a Supporting Information [62].

## Qualitative rigour

Credibility and transferability were applied for academic rigour [40]. According to Creswell [63], credibility refers to the believability and trustworthiness of a research findings. In this regard, the first author encouraged the participants to narrate their stories at their own pace while assuring them that all their stories were valid [64]. The repeated listening to the audio-recorded interviews and iterative reading of transcripts to verify 'internal consistency' was done throughout data analysis [65]. In validating the narratives, member-checking was conducted, with five participants confirming that the transcripts and their interpretations were a true reflection of what they narrated [46]. In ensuring transferability (i.e., qualitative validity, the degree to which the findings can be transferred to other contexts with other respondents), a thick rich description of the in-depth interview processes and presented findings with supporting quotes are provided. The presented themes were developed from the participants' narratives of their real-life traumatic experiences of the onset of TI during their rape ordeals. Other processes of the research were described to enable replication. To maintain accuracy and validity, and to avoid conflicting aspects, the transcriptions were reiteratively cross-checked with the audio-recorded interviews by both authors, which assisted the authors to define and refine the themes as findings.

## Findings

### Demographic characteristics of participants

Five themes emerged from the analysis of the narrated stories from the 13 participants: (1) Onset of tonic immobility prior to rape due to perceived imminent danger; (2) Onset of tonic

immobility as self-protection mechanism from further harm; (3) self-loathing as meaning attached to tonic immobility; (4) suicidal ideations as meaning attached to tonic immobility, and (5) Divine intervention as meaning attached to tonic immobility.

**Theme 1. Onset of tonic immobility prior to rape due to perceived imminent danger.**
This theme emerged from participants' narratives of experiencing symptoms suggestive of TI when they perceived threats of bodily harm from the assailants who were equipped with various weapons and threatened victims into submission to being raped. Bodily changes representing TI were experienced as a response to extreme fear from sensing imminent danger to harm from such weapons, as expressed in the following extracts:

> It was still early in the evening that day after work, on my way to the bus stop, when four men approached me from nowhere. One of them showed me a big/ shinning knife that he had with him, while another one of them asked me to follow them. Immediately when I saw that knife, I was shocked and froze there and then. My whole body just went weak. I did not know what to do. I did not scream; I could not scream; I was too terrified. I just followed them; I do not know how. . .I blocked my mind. I think they all raped me. . ." **(P1, 26–30).**

> I am a hotel receptionist. On this night, a guest called that he had a problem with his room, so I went to assist him. When I entered his room for him to show me what was wrong so I could help him, he grabbed me from behind, held me tightly, and pushed me further into the room. He threatened me with a gun and a knife, and said if I did not cooperate with him, he would use both on me. My body went cold. I was confused, afraid, and just felt very weak and could not move; my knees could not carry me. He pushed me on to the bed. That was when he raped me. I just looked at him and could not even scream or fight him off. **(P2, 26–30).**

> As a food vendor, I sell food around. Since my husband lost his job, I have been selling cooked food. So, this man I have known for a while called me to buy food from me. He asked me to get a plate from his kitchen, and that is where he attacked me. He brought out a kitchen knife from a kitchen drawer and pointed it at me. It was unbelievable, I wish I did something to stop him, but I could not. I was rooted there. I could not move or say anything. I obeyed all his commands. He raped me. I was weak, shocked, and ashamed, but I did not feel anything about what he did to me. **(P3, 31–35).**

> They pushed me down. It seemed like a long while before they started ripping off my clothes, tearing my bra, and doing all sorts of things to my body including running a razor blade over my naked body that I realised this was the end for me. So, I was terribly scared by then, and s unable to shout. I could not even move my body. I could not even struggle; I just lay there on the ground and closed my eyes. Each one of them raped me after the other and I was just lying there like a rod. **(P4, 26–30).**

**Theme 2: Onset of tonic immobility as self-protection mechanism from further harm.**
This theme derived from the data suggestive of the onset of TI following initial acts of aggression and violence towards the women who did not see the likelihood of escaping. Their bodies responded by going into TI as self-protection mechanism from the intensity of further attacks and physical injuries, as indicated in the excerpts:

> My friend's boyfriend was to take me home, but he first asked me for sex, and I refused. He got violent and started hitting me and threatening to throw me out of his room. This was around 2 am. I knew that I was stuck there. He started beating me. I first tried to fight him, but he did

*not stop beating me. Suddenly I could not move, my arms could not fight him. I could not scream either. I could no longer move but knew what he was doing to me. He pulled down my clothes, ripped my underwear and did it. (P5, 26–30).*

*As he was approaching me, I was retreating backward against the wall, and become deeply scared. I tried to run but, where would I run to? He pounced on me and was hitting me everywhere. He pushed me onto the bed with force, was on top of me, tore my clothes with such anger and a fierce look in his eyes. I lay frozen with fear on the bed where he pushed me. He stopped hitting me, maybe thinking I was dead, but he continued and raped me repeatedly (P6, 31–35).*

*On my way from my granny's home, I passed by three men whom I thought were talking among themselves. Immediately I passed them, I felt a hard slap on my back. This was without provocation. The other two joined in beating me. I was overwhelmed and did not struggle with them. They tossed me about before throwing me on the bare ground; and raped me, one after the other. After a while, the beating stopped, I do not know when because I have never been that scared before. I did not try to defend myself; I was shattered. I knew what was happening, but I just did not react because of my fear of them. (P7, 16–20).*

*He had his way; I was raped. I was not able to shout or do anything, I was just in shock and so confused. I could not believe what was happening at that time to me and by this person. He was a friend. My body felt heavy and motionless on the rug where I had collapsed. At some point, he stopped being violent. It was just a lot for me to process. . . I remained there, tuned off. I was shaking, sweating, very confused, and crying. I was so lost and confused. (P8, 26–30).*

**Theme 3: self-loathing as meaning attached to TI experience.** This theme was framed from the data expressing the participants' narratives of the meaning and emotions they later experienced towards themselves for failing to do something to protect themselves from their attacker. These emotions included self-shame, loss of self-respect, humiliation, and guilt feeling, as indicated by some of the comments below:

*After it (rape), I just felt as if the whole world knew that I gave in to that without a fight or a cry for help. I felt that my self-worth was reduced, as if everyone knew what had happened to me. And that kind of made me ashamed of myself, lose my confidence, and feel unworthy. I had let myself down, obviously.* **(P3, 31–35).**

*I lost it all: lost my self-respect and career as I could not avoid what happened to me. I see myself now as a second-class citizen, no longer worthy to be a human being. I have nothing to offer; my dignity is gone from me* **(6, 31–35).**

*When I remember how I was so helpless and let it happen, the whole thing makes me feel less and less of a human being. It has lowered my self-esteem. I do not even express myself in public anymore. I have been very withdrawn since the incident and now live inside my shadow and my shell. (P9, 21–25).*

*I feel useless, knowing I could have fought back. I had been beaten and because that's basically what happened, I now feel like a complete loser. (P5, 26–30).*

*Feeling about myself, I will say that after what happened, I feel dirty, ashamed, unworthy, and that something valuable had been taken away out of me. So, I did not know whether I was just ashamed to tell you the truth, but I felt dirty, and I hated everything about myself after that (P3, 31–35).*

*I felt bad about myself and the bad things that kept happening to me. And this is not the first time that I've been raped. The people around me always had a bad impression of me, which I hate. I hate myself and I do not want to be a burden to anyone.* **(P10, 26–30).**

**Theme 4. Suicidal ideation as meaning attached to tonic immobility.** This theme emerged from participants' narratives about wanting to die due to not being able to defend themselves during the rape, recriminating the fact that they made no effort to protect themselves, as indicated in the following extracts:

*When I think and remember that I just lay there and did nothing to defend myself, I feel like killing myself because it is so humiliating. How can I even mention that something like that happened to me, and I did nothing? What am I living for? **(P11, 16–20).***

*Whenever I think about how I just lay down there and failed to protect myself, I feel like I am not worthy of living. I feel like committing suicide. Sometimes I feel like going away forever. Moreover, it has changed everything about me, I am not as free as I used to be, I have become withdrawn, and very sad. So why must I continue to live? **(P6, 31–35).***

*Not being able to defend myself from the hands of my attacker was the experience that made me have a death wish. It is a terrible experience. After that, I do not see myself as a complete person again; it is like I allowed something to be stolen from me and did not fight for it. I feel like a total mess and cease to exist in my mind after that experience. **(P8, 26–30).***

*Because the first person who sexually abused me as a young girl made me believe that anything that was happening to me was already written and signed by God and not a mistake. With the recent rape, I felt deep hate towards myself. I told myself that it was better to die, that I was going to kill myself because I felt I would not be able to heal from all my pains (**P12, 26–30).***

**Theme 5: Divine intervention as the meaning made of tonic immobility.** This theme emerged from the participants' narratives which infer making meaning of their experiences from spiritual perspectives as indicated from their comments:

*Even though lying there paralysed and not able to do anything was bad and terrible, it proved to me that God loves me. I believe that it was God that was working for me. He confused my assailants to think I was dead so that they could stop the rape. I thank God. The men who assaulted me threatened to deal with me, but God confused them. They left me after raping me, they could have killed me; I give God the glory. He is truly a living God.* **(P4, 26–30).**

*After the second experience of a rape that I could not prevent by fighting off the attacker, I could not stand the shame and reminders each time I got home. For this, I decided to relocate to another state from Lagos. It was painful. It was humiliating that I did nothing. But by the grace of God, it helped me decide finally to leave Lagos for another state to continue with my life.* **(P9, 21–25).**

*I could not fight off my attacker, but something made me stronger, afterward. I got up and realised that I could walk from there; something somehow gave me my power back. I walked away from the scene and looked for help. God indeed helped me.* **(P5, 26–30).**

*For days I could not go out. I cancelled all the church activities I had planned for that week, asking myself, "What kind of church activities will I carry out when I don't feel God's presence around me anymore? Where was He while I was being raped?" But again, I sometimes think*

*that maybe it was God's intervention that made me to be still and do nothing in order to protect me.* **(P13, 26–30).**

## Discussion

The aim of this study was to explore and describe the onset and meaning attached to the experience of TI as a phenomenon among African raped women in Nigeria. Our findings reveal that the age range of participants [Table 1] in the present study closely resemble those of the participants of a study by Moller et al. [4].

TI is a complex phenomenon that occurs mostly before rape in humans once they perceive extreme danger and has many ramifications for the victims. Although TI is an established physiologic reaction due to severe trauma, such as rape, individuals respond uniquely in its manifestations, the main manifestation being paralysis. In addition, victims appraise the situation and attempt to make meaning of their experiences. Understanding the complexities suggestive of TI is critical developing appropriate interventions for the management of victims of rape.

### Theme 1. Onset of tonic immobility prior to rape due to perceived imminent danger

The findings on the onset of TI prior to rape was due to the perceived imminent danger as a result of the presence of weapons and supports the findings about their use in rape cases. Perpetrators have been reported to use weapons to instil fear, assert control, and prevent victims from any attempt at resistance [66,67]. These findings further confirm the report by [68] that weapons use during rape was common in Africa.

The accompanying feeling of inescapability ignites a TI like state [43] when the attacker physically confronts their victim. Participants narrated how they were frozen with fear, their bodies going limp (P1, 26–30; P2, 26–30; P3,31–35), characteristics that agree with Marks' definition of TI [69]. Being confronted by an armed attacker created extreme fear in the victims due to the anticipated consequences of being attacked and injured by someone who was clearly physically stronger than they were and had every intention of harming them [44,70,71]. The participants' narratives aligned with [66], that weapon use during SV was common to instil

**Table 1. Characteristics of participants.**

| Participant no | Age range | Abuser |
|---|---|---|
| P1 | 26–30 | Male neighbors |
| P2 | 26–30 | Hotel guest |
| P3 | 31–35 | Acquaintance |
| P4 | 26–30 | Armed robbers |
| P5 | 26–30 | Boyfriend |
| P6 | 31–35 | Pastor |
| P7 | 16–20 | Unknown men |
| P8 | 26–30 | Acquaintance |
| P9 | 21–25 | Tailoring Instructor |
| P10 | 26–30 | Co-worker |
| P11 | 16–20 | Male neighbors |
| P12 | 26–30 | Male neighbor |
| P13 | 26–30 | Pastor |

fear, assert control and prevent victims from any attempt of resistance, fear being significant in the induction of TI [4,72]. This study affirms [68] report that weapon use during rape was common in Africa, which could be a health concern in Nigeria, as traumatic situations suggestive of TI abound and are linked to PSTD [73].

There are occasions where TI occurs immediately after rape, which [73] describe as a severe and persistent traumatic state that occurs after a traumatic event. It is explained as a feeling of 'flopping down', that is, when the victim falls to the ground and just cries [74]. Narratives like this are central to backing up theories and explaining why victims of rape are said to be unable to defend themselves but can explain in detail what had happened. These behaviour manifestations often confound first responders, police, lawyers, and health practitioners, due to lack of understanding of TI, who they fail to accept how a victim is able relate the horrific experience of rape but unable to explain why they could not do anything to resist it. Similar to the findings of [74], the findings from this theme also clearly show the need to further inform individuals about all responses to TI trauma and to indicate that it is both a common and natural reaction to SV.

## Theme 2. Onset of tonic immobility as self-protection mechanism from further harm

TI has also been referred to as 'death feigning', which is regarded as an adaptive anti-predator behaviour [43,75,76], and a self-protection mechanism from further harm, given that they had already been physically assaulted and did not know when it would end [77,78]. Similar narratives were reported in the present study, some participants (P5 26–30; P6, 31–35; P7, 16–20; & P8 26–30) expressed halt catatonia to their advantage. The scientists mentioned further that that feigning death might deceive an attacker into believing that the victim is disgusting and giving up. Being paralysed was more accurately recognised as a self-protective stress reaction outside the volition of the victim [43]. Ford and partners [79] endorsed the assertion, stating that in a severe trauma situation, with no other viable escape option, the body's energies are redirected toward staying alive, hence inactivity. According to [75,76], TI at this point could protect victims from increased injury, force, severity of the attack and could provide a means of escape to victims. However, [80] questioned the protective nature of TI, insisting that some victims get hurt, despite 'feigning their death'. Our findings do not indicate if any additional harm happened during the rape as the victim lay helpless, but there is a strong indication that perpetration happens even when the victims were 'frozen' or motionless, thus indicating that such 'paralysis' did not stop the perpetrators from completing their acts of sexually assaulting the victims because of their paralysed state.

## Theme 3: Self-loathing as meaning attached to tonic immobility

Our findings revealed self-loathing and self-blaming among participants who experienced TI due to their inability to protect themselves from being raped, fight the perpetrators, or scream to be rescued having been incapacitated by the paralysis. As the participants had no control over their ability to defend themselves from the rapist due to their involuntary paralysis, they took this failure as personal, and developed self-loathing. Self-loathing on its own is not a disorder, but it is one of several possible symptoms of depression. The participants did not understand that their body can become physiologically stressed under extreme fear to the point where they are rendered incapable of moving or defending themselves, or calling out, restrained thereby rendering them completely vulnerable to being violated [16,81].Victims therefore become angry and hate themselves for their inactivity, feeling that they allowed the rape to happen, which results in the emotional destruction of their self-pride, dignity and

sense of integrity being incalculable, and for which they feel responsible. Similarly, [82] corroborated the view of emotional self-destruction among participants due to their misconception of their physical inactivity during rape. One of the worst emotions sexually abused persons experience is the sense of worthlessness and self-hatred, which is often accompanied by the feelings of guilt and shame [43,83].

In another study, rape survivors who experienced TI reported to have felt guilty, had negative self-assessment of themselves and be emotional withdrawn [20]. Similar conclusion was made regarding victims 'experiences of TI being associated with the risk for developing self-loathing from self-blame [83]. Lloyd et al. [20] referred to the emotions as self-loathing, which is a state of diminished sense of well-being and a trauma-related altered state. For self-loathing to occur, victims would have been involved in extreme self-blame, self-criticism, self-demeaning, and insult to oneself [84]. This might have led to the ordeal of this participant who felt that it was her fault that she was raped, and so narrated. . . *"I have been very withdrawn since the incident and now live inside my shadow and my shell". (P9, 21–25, raped by a tailoring instructor)*. This participant was reacting this way because she was blaming herself for being raped. Jordan [85] and Rousseau, [86] warned that such self-blame is very damaging and could cause the victim more trauma.

As self-hate has been found to predict suicide ideation [87], it is essential for health practitioners to understand self-loathing as the feeling that survivors attach to experiencing TI. This will enable those providing care and support to specifically look for such emotions, understand their origin and reassure them that they are not to blame for what happened and how they responded during the incidence. Information regarding the onset of TI and its effects on survivors of rape should be formally taught to all health professionals during their undergraduate training and as part of continuing education programmes.

Regarding meaning attached to the experience of TI, participants did not attach any meaning to the experience of TI. According to Park, [38], *meaning is made* only when people have a different view of the situation and have altered their beliefs and goals to reclaim consistency among them. This could be so because, participants were recruited only after twelve weeks of the experience of TI. TI is a behavioural response and to alter behaviour requires a considerable time. At present, studies where meanings have been made from SA, participants were abused as children and were only asked to make meanings to it as adults [88].

## Theme 4: Suicidal ideation as meaning of tonic immobility

Identifying suicidal ideation (SI) as the consequence of experiencing TI became one of the major concerns in our study. Four participants who narrated this were immediately referred to the psychologist of their choice among the two who were on standby for additional follow-up care. Thoughts of suicide in response to TI is consistent with other findings on the victim's experiences of SV trauma [89–93]. SI following the experience of TI is often associated with guilt, shame, and self-blame, and is like other findings that note that they intensify notions of suicide [90]. Magalhaes et al. [94]'s observation that prolonged TI is associated with PTSD symptoms in individuals who had experienced trauma agree with Moller et al's study [4], which showed that 70% of the 298 sexually assaulted women experienced or developed psychological complications. Conversely, Brewin and Partners [95] contended that only a small percentage of victims met the criteria for trauma-related stress disorders. This may explain why some studies report that suicide is often considered by many rape survivors [96], but not all persons who have suicidal ideations attempt suicide [97]. However, despite these findings even a low rate of suicidal thoughts among victims is critical enough to be alarmed.

Based on these findings, we recommend that health practitioners be encouraged to assess for suicide ideation during follow-up visits among all rape survivors, specifically those who

reported to have experienced TI and show signs of self-loathing. It is therefore crucial that all rape victims who report to have experienced TI during their ordeal to be assessed for early identification of SI as a global public health concern (ICD 10-CM) and the second leading cause of death worldwide [98,99]. This is especially so in Nigeria, where a study recently reported that the pooled prevalence of suicidal ideation was 9.8% and pooled suicide attempts was 16.2% annually [100]. Although the research was on HIV, influencers of SIs, the SI influencers of the study were similar to psychological symptoms experienced by the participants in this study. Moreover, [101] declare that SIs are a warning signs for and regulate the prediction of completed suicides. The findings from this theme show the need to further educate individuals on appropriate responses to trauma, and to highlight that the need for a strategic responsive intervention mechanism that will improve care and support and counter the development of psychological complications from TI among individuals. According to the meaning making model following trauma by [35], after trauma, participants are expected to be involved in reasoning evaluation of the situation against their held beliefs or values. But the outcome of their evaluation being SI means they could not find any favorable meaning to tie their emotions after the abuse to, hence inconsistencies waiting to be resolved and SI in the interim.

## Theme 5: Divine intervention as meaning made of tonic immobility

The findings related to understanding the onset of TI as a divine intervention reveal the importance of finding meaning spiritually among survivors of rape and how they cope with the unexplainable, because some individuals in Nigeria believe rape to be a spiritual phenomenon [102,103].

Park, [35] explained that usually, the purpose of meanings made are to change the meaning of a situation to a favourable to promote growth. Spirituality is an important factor towards meaning making and healing among victims of SA [23,26,104] as participants often took the situation as being allowed by God. Further, some participants in the present study associated the inability of physical movement of their body as the power and tranquillity from a Divine Source that manifested to help and protect them from a worse fate during their ordeal [105]. This was supported as shown below by an extract from this study: "*It proved to me that God loved me. I believe that it was God that confused my assailant to think I was dead*" (P9 21–25). She found meaning in her TI state and attributed it to God, and this changed the very meaning of the distressful meaning of experience of TI [106]. According to context of trauma, this is known as being triumphant despite devastation that is the ability to convert the view of a personal ordeal to a personal achievement [107]. Thus, the meaning made here is seen as an embodiment to a new understanding that will ultimately, promote well-being [38,108]. Masten, [109], alludes this new understanding to resilience which according to him, showed the ability to adapt to and recover from adverse circumstances that enable one to return to improved levels of functioning. On the other hand, another participant who felt that divine intervention should have prevented the incidence from occurring in the first place, rather asked "Where was He (God) while I was being raped?" (P13, 26–30), suggesting that if there was a God that this event should not have happened. Cummings and Pargament, [110] acknowledged that some individuals may question God, as P13 did, hence indicating negative coping and meaning that meaning was not made here.

Messina-Dysert [96] endorsed that SA can have a damaging effect on the spiritual health of women thereby testing their faith and displaying their frustration with God. Spirituality often weighs heavily in individual's efforts to deal with serious trauma and a frequently used resource for people coping with a variation of stressful and traumatic experiences [23,38]. Conversely, a northern tribe in Nigeria considered psychological symptoms to mean 'destiny',

believing that the symptoms are ordained by God [103]. Often, according to [111], spiritual meaning making gives a source of comfort and intelligibility during a difficult time. Such individuals may not consider medical management when experiencing psychological problems but rather believe that it is God's wish for them, and the danger with this is the transition to severe psychological stress may be missed. Despite the differences in understanding the role of the divine in the rape experience, and the consequences for the value of spirituality in the recovery process and spiritual health, we recommend that health practitioners acknowledge the survivors' spiritual beliefs in their provision of holistic care by referral.

## Strengths/Limitations

The study has strengths as well as limitations. The major strength to the best of our knowledge, this is the first study of its kind in Nigeria, it was Afrocentric, and contend that it will promote further research interest in this area and would add to decisions that would inform the development of appropriate interventions for good mental health of affected women and commiserate legal outcomes for survivors of rape. Although a small sample size was used, the study findings are informed by women's voices and thus provide authentic evidence of the onset of TI among the participants. In addition to the strength of the study is the findings on meaning making which are scarce in sexual abuse studies, this study may open interest in this area as much as it is at present with child sexual abuse studies.

The obvious limitations of this study were conducting a narrative approach using English and the Nigerian Pidgin language instead of the range of indigenous Nigerian languages. Another limitation to this study is the fact that there is little or no research on this topic and could not form part of the present study. The recruitment of participants was challenging as such we even had to change the original recruitment plan from poster to direct approach. With over 350 known spoken languages, and Lagos as a cosmopolitan city, it could have difficult for the first author to be familiar with all spoken languages, hence the use of these two widely spoken languages. While some participants could not fully express their experience of TI, they were assisted to use body language and signs to express themselves, which may have resulted in some sensitive data being missed in translation.

Findings made on meanings made may have been different probably if study was a longitudinal type or if abused women had had the experience of TI for more than twelve weeks before data collection. This is because, change is a gradual process and the phenomenon under study affects emotions and behaviour all of which change with time.

## Contribution to knowledge and recommendations for future research

The present study contributes to knowledge about the onset of TI by identifying experiential triggers of TI among African participants. As first research, it provides valuable base line data for future research on TI in Nigeria and West of Africa. Some findings validate the effects of TI from rape and show meaning made by Africans to the outcome of TI, which raises the impossibility of uniform intervention algorithm across the globe and measures for identifying TI. First responders such as nurses, must learn to appropriately deal with rape victims by being trauma aware. To strengthen the present findings, it would be beneficial to conduct that future research consider a larger study and across all age groups since rape cuts across all ages to have more options for responsive treatment plans. Knowledge generated from the narrations of the onset of TI might provide insight into understanding the reactions of victims to rape, believability of rape reports, and why some victims think of suicide. It would also be beneficial that future consider the grounded theory to TI study to originate principles of care from it.

Quantitative studies should therefore endeavour to address the current gaps regarding phenomenon development in an attempt to achieve more knowledge, understanding and improved therapeutic practice; in addition, provide other bases for much subsequent future research on this subject. We also recommend that researchers conducting sensitive researcher to consider the use of a mother tongue for in-depth interviews, or to invest in having appropriately trained and qualified language translators if there is a language difference. In addition, findings from these studies could facilitate the development of policies and interventions to alter rape-supportive attitudes and beliefs and enhance believability of reports of rape. Lastly, as we were only able to engage five of the 13 survivors for member checking, it would be worthwhile to attempt to apply on the spot member checking, going further in sensitive research, especially where stigma is high.

## Conclusion

There is a strong likelihood that TI is not an uncommon experience for rape victims, but that in the absence of research and public reporting on the condition, the shame and self-blame associated with it will haunt many women. It will not only affect the quality of life of millions of these women who but their emotional wellbeing, sense of self-worth and relationships with other people, who may or may not believe and support them. In societies where justice may not run its full course for those traumatised by rape, salvaging their self-worth and esteem becomes essential to prevent women from suffering from a double injustice when all the systems fail them due to lack of knowledge.

## Supporting information

**S1 File.**
(PDF)

**S2 File.**
(DOCX)

## Acknowledgments

We acknowledge rape victims and survivors all over the world, especially participants in this study, Nigeria, who shared their experiences with us for this study. We also acknowledge the governmental and one NGO post rape care and management facilities for granting us access to their facilities and most especially those specific staff that were assigned to assist with the initial recruitment process.

## Author Contributions

**Conceptualization:** Dooshima Dorothy Gbahabo, Sinegugu Evidence Duma.

**Data curation:** Dooshima Dorothy Gbahabo, Sinegugu Evidence Duma.

**Formal analysis:** Dooshima Dorothy Gbahabo, Sinegugu Evidence Duma.

**Funding acquisition:** Sinegugu Evidence Duma.

**Investigation:** Dooshima Dorothy Gbahabo.

**Methodology:** Dooshima Dorothy Gbahabo, Sinegugu Evidence Duma.

**Project administration:** Dooshima Dorothy Gbahabo, Sinegugu Evidence Duma.

**Resources:** Dooshima Dorothy Gbahabo, Sinegugu Evidence Duma.

**Supervision:** Sinegugu Evidence Duma.

**Validation:** Sinegugu Evidence Duma.

**Visualization:** Dooshima Dorothy Gbahabo, Sinegugu Evidence Duma.

**Writing – original draft:** Dooshima Dorothy Gbahabo.

**Writing – review & editing:** Dooshima Dorothy Gbahabo, Sinegugu Evidence Duma.

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
