## [Decision Letter · Decision Letter 0]

13 Apr 2023

PONE-D-22-32334...I did not scream. I could not; I was terrified.  I just followed them...I blocked my mind. Then they all raped me: A narrative inquiry on the onset of tonic immobility among women rape victims in Nigeria.PLOS ONE

Dear Dr. Dooshima Dorothy Gbahabo

Thank you for submitting your manuscript to PLOS ONE. After careful consideration, we feel that it has merit but does not fully meet PLOS ONE’s publication criteria as it currently stands. Therefore, we invite you to submit a revised version of the manuscript that addresses the points raised during the review process.

Pleas follow revisor's suggestions.

We look forward to receiving your revised manuscript.

Kind regards,

Kornelia Zaręba, MD

Academic Editor

PLOS ONE

Journal Requirements:

3. Please include a complete copy of PLOS’ questionnaire on inclusivity in global research in your revised manuscript. Our policy for research in this area aims to improve transparency in the reporting of research performed outside of researchers’ own country or community. The policy applies to researchers who have travelled to a different country to conduct research, research with Indigenous populations or their lands, and research on cultural artefacts. The questionnaire can also be requested at the journal’s discretion for any other submissions, even if these conditions are not met.  Please find more information on the policy and a link to download a blank copy of the questionnaire here: https://journals.plos.org/plosone/s/best-practices-in-research-reporting. Please upload a completed version of your questionnaire as Supporting Information when you resubmit your manuscript.

4. Please amend your current ethics statement to address the following concerns:

a) Did participants provide their written or verbal informed consent to participate in this study?

Reviewers' comments:

Reviewer's Responses to Questions

**Comments to the Author**

1. Is the manuscript technically sound, and do the data support the conclusions?

Reviewer #1: Yes

Reviewer #2: Yes

2. Has the statistical analysis been performed appropriately and rigorously? 

Reviewer #1: Yes

Reviewer #2: I Don't Know

3. Have the authors made all data underlying the findings in their manuscript fully available?

Reviewer #1: Yes

Reviewer #2: Yes

4. Is the manuscript presented in an intelligible fashion and written in standard English?

Reviewer #1: Yes

Reviewer #2: Yes

5. Review Comments to the Author

Reviewer #1: Dear Authors, thank you for the opportunity to review your interesting and feministic, human-oriented qualitative study. The most important value of your research are related to:

1. Depathologization of TI. Thinking in terms of psychiatric conditions might be leading to self stigma and might be another trigger of self-loathing in many cultures.

2. Descriptive character of the study might lead to better understanding of TI phenomenon in clinical practice.

Reviewer #2: I appreciate your work on such sensitive topic, I have realized that there are few highlighted paragraphs (lines 221-235) and (lines 517-547), is it been revised?

I also suggest creating a table for your sample characteristics; present their ages, raped person, or any other data.

you have mentioned in your introduction that there are limited knowledge about TI; I think it is important to suggest quantitative studies in the future about the same topic for example "prevalence of TI among rape victims" to find out the about the prevalence and to work forward accordingly.

6. PLOS authors have the option to publish the peer review history of their article (what does this mean?). If published, this will include your full peer review and any attached files.

Reviewer #1: No

Reviewer #2: **Yes: **Nazdar Qudrat Abas

---

## [Author Response · Author response to Decision Letter 0]

31 Aug 2023

Response to Reviewers 

Subject: PLOS ONE Decision: Revision required [PONE-D-22-32334] - [EMID:7687612a4ba479bd]

...I did not scream. I could not; I was terrified. I just followed them...I blocked my mind. Then they all raped me: A narrative inquiry on the onset of tonic immobility among women rape victims in Nigeria.

Author’s Response: 

Done

Author’s Response: 

Done

Author’s Response: 

Done

Author’s Response: 

Not applicable

Author’s Response: 

Not applicable

We look forward to receiving your revised manuscript.

Kind regards,

Kornelia Zaręba, MD

Academic Editor

PLOS ONE

Author’s Response: 

Thank you.

Journal Requirements:

Author’s Response: 

We have scrutinized our manuscript to the best of our knowledge to ensure compliance of our manuscript with PLOS ONE’s style requirements.

Author’s Response: Comments 2 and 3 are the same.

3. Please include a complete copy of PLOS’ questionnaire on inclusivity in global research in your revised manuscript. Our policy for research in this area aims to improve transparency in the reporting of research performed outside of researchers’ own country or community. The policy applies to researchers who have travelled to a different country to conduct research, research with Indigenous populations or their lands, and research on cultural artefacts. The questionnaire can also be requested at the journal’s discretion for any other submissions, even if these conditions are not met. Please find more information on the policy and a link to download a blank copy of the questionnaire here: https://journals.plos.org/plosone/s/best-practices-in-research-reporting. Please upload a completed version of your questionnaire as Supporting Information when you resubmit your manuscript.

Author’s Response: 

A completed copy of PLOS’ questionnaire on inclusivity in global research is uploaded as Supporting Information I.

4. Please amend your current ethics statement to address the following concerns:

a) Did participants provide their written or verbal informed consent to participate in this study?

Author’s Response: Amended. Refer Pages 8 and 10

Yes, ethics committees approved the proposal that prospective participants who were willing to participate in the study would sign consent forms.

Author’s Response: This study’s minimal underlying data set has been uploaded as Supporting Information II

Author’s Response: Noted

Author’s Response: 

i. Reference list number 49

Shrestha, B., & Dunn, L. (2019). The declaration of Helsinki on medical research involving human subjects: a review of seventh revision. Journal of Nepal Health Research Council, 17(4), 548-552 

Below is the former reference S/no 49, that has been replaced with the one above for recency.

‘Association WM. WMA Declaration of Helsinki—ethical principles for medical research involving human subjects. 2013’. 

ii. This reference ‘Ochberg F. Post-traumatic therapy and victims of violence…’ was left in the reference list as S/no 86 in error. It has been deleted.

iii. Reference no 62 was inserted [PLOS Questionnaire on Inclusivity in Global Research] newly to accommodate the journal requirement S/no 2. With this, reference serial numbers had to change. Former serial no 62, became 63 and this continued until we got the last serial number 113.

iv. Frankl VE. Man’s search for meaning. Rider, S/no 109 was replaced with a more appropriate author - Pattakos, A. (2010). Prisoners of our thoughts: Viktor Frankl's principles for discovering meaning in life and work. Berrett-Koehler Publishers.

Reviewers' comments:

Comments to the Author

1. Is the manuscript technically sound, and do the data support the conclusions?

Reviewer #1: Yes

Reviewer #2: Yes

2. Has the statistical analysis been performed appropriately and rigorously?

Reviewer #1: Yes

Reviewer #2: I Don't Know

3. Have the authors made all data underlying the findings in their manuscript fully available?

The PLOS Data policy requires authors to make all data underlying the findings described in their manuscript fully available without restriction, with rare exceptions (please refer to the Data Availability Statement in the manuscript PDF file). The data should be provided as part of the manuscript or its supporting information or deposited to a public repository. For example, in addition to summary statistics, the data points behind means, medians, and variance measures should be available. If there are restrictions on publicly sharing data—e.g., participant privacy or use of data from a third party—those must be specified.

Reviewer #1: Yes

Reviewer #2: Yes

4. Is the manuscript presented in an intelligible fashion and written in standard English?

Reviewer #1: Yes

Reviewer #2: Yes

5. Review Comments to the Author

Reviewer #1: Dear Authors, thank you for the opportunity to review your interesting and feministic, human-oriented qualitative study. The most important value of your research are related to:

1. Depathologization of TI. Thinking in terms of psychiatric conditions might be leading to self-stigma and might be another trigger of self-loathing in many cultures.

Author’s Response: 

Yes, while agreeing with you the complications relating to self-stigma and self-loathing [as our study also deduced], TI is an emerging phenomenon [Gama et al., 2022], so the term ‘depathologization’ may not particularly apply to it. The phenomenon is regarded as a behavioral state that is characterized by behavioral symptoms rather than a disease (Edelaar et al., 2012; Miyatake et al. 2009). Thus, it has largely been associated with posttraumatic stress symptoms rather than stemming from a disease entity. [Gama et al., 2022; Magalhaes et al., 2021; Moller et al., 2017]. This is why it was considered the best option not to include participants that in addition were already mentally challenged and those who chose to opt-out were respected [these possibly opted out due to sociocultural factors]. In addition, the findings of the study could serve for the common good, especially in Nigeria where the researchers believed that this study might be the first of its kind. [References are on page 7].

2. Descriptive character of the study might lead to better understanding of TI phenomenon in clinical practice.

Author’s Response: 

Thank you. It is hoped that the background to the study and the methods section achieves this.

Reviewer #2: I appreciate your work on such a sensitive topic, I have realized that there are few highlighted paragraphs (lines 221-235) and (lines 517-547), is it been revised?

Author’s Response: I checked my copy of the submitted manuscript but could not find the highlighted sections you referred me to. Kindly, I would appreciate any other guide to help me attend to the comments. Thank you.

I also suggest creating a table for your sample characteristics; present their ages, raped person, or any other data. See page 12.

Author’s Response: 

Done, table created as Table I. However, participants’ descriptors were limited to Participant number, age range, and type of rape to shroud the identity of the victims.

You have mentioned in your introduction that there are limited knowledge about TI; I think it is important to suggest quantitative studies in the future about the same topic for example "prevalence of TI among rape victims" to find out the about the prevalence and to work forward accordingly.

Author’s Response: 

Done, refer along line 645, page 28. 

6. PLOS authors have the option to publish the peer review history of their article (what does this mean?). If published, this will include your full peer review and any attached files.

Author’s Response: 

If you choose “no”, your identity will remain anonymous, but your review may still be made public.

Author’s Response: 

Affirmative

Do you want your identity to be public for this peer review? For information about this choice, including consent withdrawal, please see our Privacy Policy.

Reviewer #1: No

Reviewer #2: Yes: Nazdar Qudrat Abas

Author’s Response: 

Alright

Author’s Response: 

Not applicable

*References to Authors response to Reviewer 1’s comment number 1

Gallup GG, Rager DR. 1996. Tonic immobility as a model of extreme states of behavioral inhibition. In: Sanberg PR, Ossenkopp K-P, Kavaliers M, editors. Motor activity and movement disorders: research issues and applications. New Jersey (NY): Humana Press. p. 57–80

Gama, C. M. F., de Souza Junior, S., Gonçalves, R. M., Santos, E. D. C., Machado, A. V., Portugal, L. C. L., Passos, R. B. F., Erthal, F. S., Vilete, L. M. P., Mendlowicz, M. V., Berger, W., Volchan, E., de Oliveira, L., & Pereira, M. G. (2022). Tonic immobility is associated with posttraumatic stress symptoms in healthcare professionals exposed to COVID-19-related trauma. Journal of anxiety disorders, 90, 102604. https://doi.org/10.1016/j.janxdis.2022.102604

Magalhaes, A. A., Gama, C. M. F., Gonçalves, R. M., Portugal, L. C. L., David, I. A., Serpeloni, F., Wernersbach Pinto, L., Assis, S. G., Avanci, J. Q., Volchan, E., Figueira, I., Vilete, L. M. P., Luz, M. P., Berger, W., Erthal, F. S., Mendlowicz, M. V., Mocaiber, I., Pereira, M. G., & de Oliveira, L. (2021). Tonic Immobility is Associated with PTSD Symptoms in Traumatized Adolescents. Psychology research and behavior management, 14, 1359–1369. https://doi.org/10.2147/PRBM.S317343

Miyatake T, Nakayama S, Nishi Y, Nakajima S. 2009. Tonically immobilized selfish prey can survive by sacrificing others. Proc R Soc B Biol Sci. 276:2763–2767.

Möller, A., Söndergaard, H. P., & Helström, L. (2017). Tonic immobility during sexual assault–a common reaction predicting post‐traumatic stress disorder and severe depression. Acta obstetricia et gynecologica Scandinavica, 96(8), 932-938.

---

## [Decision Letter · Decision Letter 1]

30 Oct 2023

...I did not scream. I could not; I was terrified.  I just followed them...I blocked my mind. Then they all raped me: A narrative inquiry on the onset of tonic immobility among women rape victims in Nigeria.

PONE-D-22-32334R1

Dear Dr. Gbahabo,

We’re pleased to inform you that your manuscript has been judged scientifically suitable for publication and will be formally accepted for publication once it meets all outstanding technical requirements.

Kind regards,

Michal Ptaszynski, PhD

Academic Editor

PLOS ONE

Additional Editor Comments (optional):

Reviewers' comments:

Reviewer's Responses to Questions

**Comments to the Author**

1. If the authors have adequately addressed your comments raised in a previous round of review and you feel that this manuscript is now acceptable for publication, you may indicate that here to bypass the “Comments to the Author” section, enter your conflict of interest statement in the “Confidential to Editor” section, and submit your "Accept" recommendation.

Reviewer #1: All comments have been addressed

Reviewer #2: All comments have been addressed

2. Is the manuscript technically sound, and do the data support the conclusions?

Reviewer #1: Yes

Reviewer #2: Yes

3. Has the statistical analysis been performed appropriately and rigorously? 

Reviewer #1: (No Response)

Reviewer #2: I Don't Know

4. Have the authors made all data underlying the findings in their manuscript fully available?

Reviewer #1: Yes

Reviewer #2: Yes

5. Is the manuscript presented in an intelligible fashion and written in standard English?

Reviewer #1: Yes

Reviewer #2: Yes

6. Review Comments to the Author

Reviewer #1: Dear authors, all of comments were adressed. Manuscript needed to be minimally revised. Thank you! Best regards, JG

Reviewer #2: The changes has been addressed and I think you can expand your work and studying further related factors in the future. I wish you the best of luck.

7. PLOS authors have the option to publish the peer review history of their article (what does this mean?). If published, this will include your full peer review and any attached files.

Reviewer #1: No

Reviewer #2: **Yes: **Nazdar Qudrat Abas

---

## [Editor Report · Acceptance letter]

23 Jan 2024

PONE-D-22-32334R1 

PLOS ONE

Dear Dr. Gbahabo, 

I'm pleased to inform you that your manuscript has been deemed suitable for publication in PLOS ONE. Congratulations! Your manuscript is now being handed over to our production team.

Kind regards, 

on behalf of

Dr. Michal Ptaszynski 

Academic Editor

PLOS ONE